# Causal Effect Inference with Deep Latent-Variable Models

**Christos Louizos**
University of Amsterdam
TNO Intelligent Imaging
c.louizos@uva.nl

**Uri Shalit**
New York University
CIMS
uas1@nyu.edu

**Joris Mooij**
University of Amsterdam
j.m.mooij@uva.nl

**David Sontag**
Massachusetts Institute of Technology
CSAIL & IMES
dsontag@mit.edu

**Richard Zemel**
University of Toronto
CIFAR*
zemel@cs.toronto.edu

**Max Welling**
University of Amsterdam
CIFAR*
m.welling@uva.nl

## Abstract

Learning individual-level causal effects from observational data, such as inferring the most effective medication for a specific patient, is a problem of growing importance for policy makers. The most important aspect of inferring causal effects from observational data is the handling of confounders, factors that affect both an intervention and its outcome. A carefully designed observational study attempts to measure all important confounders. However, even if one does not have direct access to all confounders, there may exist noisy and uncertain measurement of proxies for confounders. We build on recent advances in latent variable modeling to simultaneously estimate the unknown latent space summarizing the confounders and the causal effect. Our method is based on Variational Autoencoders (VAE) which follow the causal structure of inference with proxies. We show our method is significantly more robust than existing methods, and matches the state-of-the-art on previous benchmarks focused on individual treatment effects.

## 1 Introduction

Understanding the causal effect of an intervention $\mathbf{t}$ on an individual with features $\mathbf{X}$ is a fundamental problem across many domains. Examples include understanding the effect of medications on a patient's health, or of teaching methods on a student's chance of graduation. With the availability of large datasets in domains such as healthcare and education, there is much interest in developing methods for learning individual-level causal effects from observational data [42, 53, 25, 43].

The most crucial aspect of inferring causal relationships from observational data is confounding. A variable which affects both the intervention and the outcome is known as a *confounder* of the effect of the intervention on the outcome. On the one hand, if such a confounder can be measured, the standard way to account for its effect is by "controlling" for it, often through covariate adjustment or propensity score re-weighting [39]. On the the other hand, if a confounder is hidden or unmeasured, it is impossible in the general case (i.e. without further assumptions) to estimate the effect of the intervention on the outcome [40]. For example, socio-economic status can affect both the medication a patient has access to, and the patient's general health. Therefore socio-economic status acts as confounder between the medication and health outcomes, and without measuring it we cannot in

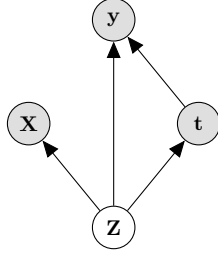

Figure 1: Example of a proxy variable. $t$ is a treatment, e.g. medication; $y$ is an outcome, e.g. mortality. $\mathbf{Z}$ is an unobserved confounder, e.g. socio-economic status; and $\mathbf{X}$ is noisy views on the hidden confounder $\mathbf{Z}$, say income in the last year and place of residence.

general isolate the causal effect of medications on health measures. Henceforth we will denote observed potential confounders[2] by $\mathbf{X}$, and unobserved confounders by $\mathbf{Z}$.

In most real-world observational studies we cannot hope to measure all possible confounders. For example, in many studies we cannot measure variables such as personal preferences or most genetic and environmental factors. An extremely common practice in these cases is to rely on so-called "proxy variables" [38, 6, 36, Ch. 11]. For example, we cannot measure the socio-economic status of patients directly, but we might be able to get a proxy for it by knowing their zip code and job type. One of the promises of using big-data for causal inference is the existence of myriad proxy variables for unmeasured confounders.

How should one use these proxy variables? The answer depends on the relationship between the hidden confounders, their proxies, the intervention and outcome [31, 37]. Consider for example the causal graphs in Figure 1: it's well known [20, 15, 18, 31, 41] that it is often *incorrect* to treat the proxies $\mathbf{X}$ as if they are ordinary confounders, as this would induce bias. See the Appendix for a simple example of this phenomena. The aforementioned papers give methods which are guaranteed to recover the true causal effect when proxies are observed. However, the strong guarantees these methods enjoy rely on strong assumptions. In particular, it is assumed that the hidden confounder is either categorical with known number of categories, or that the model is linear-Gaussian.

In practice, we cannot know the exact nature of the hidden confounder $\mathbf{Z}$: whether it is categorical or continuous, or if categorical how many categories it includes. Consider socio-economic status (SES) and health. Should we conceive of SES as a continuous or ordinal variable? Perhaps SES as confounder is comprised of two dimensions, the economic one (related to wealth and income) and the social one (related to education and cultural capital). $\mathbf{Z}$ might even be a mix of continuous and categorical, or be high-dimensional itself. This uncertainty makes causal inference a very hard problem even with proxies available. We propose an alternative approach to causal effect inference tailored to the *surrogate-rich* setting when many proxies are available: estimation of a latent-variable model where we simultaneously discover the hidden confounders and infer how they affect treatment and outcome. Specifically, we focus on (approximate) maximum-likelihood based methods.

Although in many cases learning latent-variable models are computationally intractable [50, 7], the machine learning community has made significant progress in the past few years developing computationally efficient algorithms for latent-variable modeling. These include methods with provable guarantees, typically based on the method-of-moments (e.g. Anandkumar et al. [4]); as well as robust, fast, heuristics such as variational autoencoders (VAEs) [27, 46], based on stochastic optimization of a variational lower bound on the likelihood, using so-called recognition networks for approximate inference.

Our paper builds upon VAEs. This has the disadvantage that little theory is currently available to justify when learning with VAEs can identify the true model. However, they have the significant advantage that they make substantially weaker assumptions about the data generating process and the structure of the hidden confounders. Since their recent introduction, VAEs have been shown to be remarkably successful in capturing latent structure across a wide-range of previously difficult problems, such as modeling images [19], volumes [24], time-series [10] and fairness [34].

We show that in the presence of noisy proxies, our method is more robust against hidden confounding, in experiments where we successively add noise to known-confounders. Towards that end we introduce a new causal inference benchmark using data about twin births and mortalities in the USA. We further show that our method is competitive on two existing causal inference benchmarks. Finally, we note that our method does not currently deal with the related problem of selection bias, and we leave this to future work.

**Related work.** Proxy variables and the challenges of using them correctly have long been considered in the causal inference literature [54, 14]. Understanding what is the best way to derive and measure possible proxy variables is an important part of many observational studies [13, 29, 55]. Recent work by Cai and Kuroki [9], Greenland and Lash [18], building on the work of Greenland and Kleinbaum [17], Selén [47], has studied conditions for causal identifiability using proxy variables. The general idea is that in many cases one should first attempt to infer the joint distribution $p(\mathbf{X}, \mathbf{Z})$ between the proxy and the hidden confounders, and then use that knowledge to adjust for the hidden confounders [55, 41, 32, 37, 12]. For the example in Figure 1, Cai and Kuroki [9], Greenland and Lash [18], Pearl [41] show that if $\mathbf{Z}$ and $\mathbf{X}$ are categorical, with $\mathbf{X}$ having at least as many categories as $\mathbf{Z}$, and with the matrix $p(\mathbf{X}, \mathbf{Z})$ being full-rank, one could identify the causal effect of $\mathbf{t}$ on $\mathbf{y}$ using a simple matrix inversion formula, an approach called "effect restoration". Conditions under which one could identify more general and complicated proxy models were recently given by [37].

## 2 Identification of causal effect

Throughout this paper we assume the causal model in Figure 1. For simplicity and compatibility with prior benchmarks we assume that the treatment $\mathbf{t}$ is binary, but our proposed method does not rely on that. We further assume that the joint distribution $p(\mathbf{Z}, \mathbf{X}, \mathbf{t}, \mathbf{y})$ of the latent confounders $\mathbf{Z}$ and the observed confounders $\mathbf{X}$ can be approximately recovered solely from the observations $(\mathbf{X}, \mathbf{t}, \mathbf{y})$. While this is impossible if the hidden confounder has no relation to the observed variables, there are many cases where this is possible, as mentioned in the introduction. For example, if $\mathbf{X}$ includes three independent views of $\mathbf{Z}$ [4, 22, 16, 2]; if $\mathbf{Z}$ is categorical and $\mathbf{X}$ is a Gaussian mixture model with components determined by $\mathbf{X}$ [5]; or if $\mathbf{Z}$ is comprised of binary variables and $\mathbf{X}$ are so-called "noisy-or" functions of $\mathbf{Z}$ [23, 8]. Recent results show that certain VAEs can recover a very large class of latent-variable models [51] as a minimizer of an optimization problem; the caveat is that the optimization process is not guaranteed to achieve the true minimum even if it is within the capacity of the model, similar to the case of classic universal approximation results for neural networks.

### 2.1 Identifying individual treatment effect

Our goal in this paper is to recover the individual treatment effect (ITE), also known as the conditional average treatment effect (CATE), of a treatment $\mathbf{t}$, as well as the average treatment effect (ATE):

$$ITE(x) := \mathbb{E}\left[\mathbf{y}|\mathbf{X} = x, do(\mathbf{t} = 1)\right] - \mathbb{E}\left[\mathbf{y}|\mathbf{X} = x, do(\mathbf{t} = 0)\right], \qquad ATE := \mathbb{E}[ITE(x)]$$

Identification in our case is an immediate result of Pearl's back-door adjustment formula [40]:

**Theorem 1.** *If we recover $p(\mathbf{Z}, \mathbf{X}, \mathbf{t}, \mathbf{y})$ then we recover the ITE under the causal model in Figure 1.*

*Proof.* We will prove that $p(\mathbf{y}|\mathbf{X}, do(\mathbf{t} = 1))$ is identifiable under the premise of the theorem. The case for $\mathbf{t} = 0$ is identical, and the expectations in the definition of ITE above readily recovered from the probability function. ATE is identified if ITE is identified. We have that:

$$p(\mathbf{y}|\mathbf{X}, do(\mathbf{t} = 1)) = \int_{\mathbf{Z}} p(\mathbf{y}|\mathbf{X}, do(\mathbf{t} = 1), \mathbf{Z}) \, p(\mathbf{Z}|\mathbf{X}, do(\mathbf{t} = 1)) \, d\mathbf{Z} \stackrel{(i)}{=}$$

$$\int_{\mathbf{Z}} p(\mathbf{y}|\mathbf{X}, \mathbf{t} = 1, \mathbf{Z}) \, p(\mathbf{Z}|\mathbf{X}) \, d\mathbf{Z}, \tag{1}$$

where equality (i) is by the rules of *do*-calculus applied to the causal graph in Figure 1 [40]. This completes the proof since the quantities in the final expression of Eq. (1) can be identified from the distribution $p(\mathbf{Z}, \mathbf{X}, \mathbf{t}, \mathbf{y})$ which we know by the Theorem's premise. $\qquad\square$

Note that the proof and the resulting estimator in Eq. (1) would be identical whether there is or there is not an edge from $\mathbf{X}$ to $\mathbf{t}$. This is because we intervene on $\mathbf{t}$. Also note that for the model in Figure 1,

**y** is independent of **X** given **Z**, and we obtain: $p\left(\mathbf{y}|\mathbf{X}, do(\mathbf{t}=1)\right) = \int_{\mathbf{Z}} p\left(\mathbf{y}|\mathbf{t}=1, \mathbf{Z}\right) p\left(\mathbf{Z}|\mathbf{X}\right) d\mathbf{Z}$. In the next section we will show how we estimate $p\left(\mathbf{Z}, \mathbf{X}, \mathbf{t}, \mathbf{y}\right)$ from observations of $(\mathbf{X}, \mathbf{t}, \mathbf{y})$.

## 3 Causal effect variational autoencoder

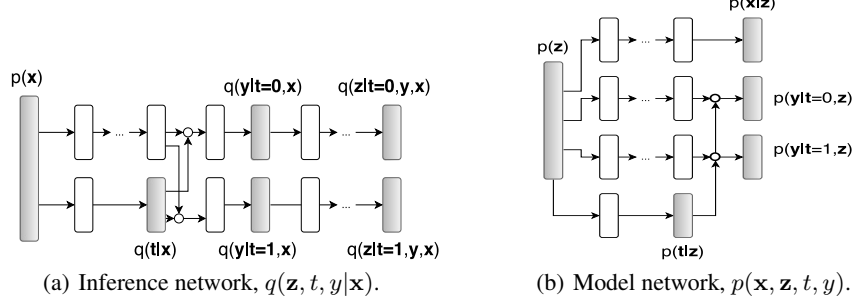

(a) Inference network, $q(\mathbf{z}, t, y|\mathbf{x})$.  (b) Model network, $p(\mathbf{x}, \mathbf{z}, t, y)$.

Figure 2: Overall architecture of the model and inference networks for the Causal Effect Variational Autoencoder (CEVAE). White nodes correspond to parametrized deterministic neural network transitions, gray nodes correspond to drawing samples from the respective distribution and white circles correspond to switching paths according to the treatment $t$.

The approach we take in this paper to the problem of learning the latent variable causal model is by using variational autoencoders [27, 46] to infer the complex non-linear relationships between **X** and $(\mathbf{Z}, \mathbf{t}, \mathbf{y})$ and approximately recover $p\left(\mathbf{Z}, \mathbf{X}, \mathbf{t}, \mathbf{y}\right)$. Recent work has dramatically increased the range and type of distributions which can be captured by VAEs [51, 45, 28]. The drawback of these methods is that because of the difficulty of guaranteeing global optima of neural net optimization, one cannot ensure that any given instance will find the true model even if it is within the model class. We believe this drawback is offset by the strong empirical performance across many domains of deep neural networks in general, and VAEs in particular. Specifically, we propose to parametrize the causal graph of Figure 1 as a latent variable model with neural net functions connecting the variables of interest. The flexible non-linear nature of neural nets will hopefully allow us to approximate well the true interactions between the treatment and its effect.

Our design choices are mostly typical for VAEs: we assume the observations factorize conditioned on the latent variables, and use an inference network [27, 46] which follows a factorization of the true posterior. For the generative model we use an architecture inspired by TARnet [48], but instead of conditioning on observations we condition on the latent variables **z**; see details below. For the following, $\mathbf{x}_i$ corresponds to an input datapoint (e.g. the feature vector of a given subject), $t_i$ corresponds to the treatment assignment, $y_i$ to the outcome of the of the particular treatment and $\mathbf{z}_i$ corresponds to the latent hidden confounder. Each of the corresponding factors is described as:

$$p(\mathbf{z}_i) = \prod_{j=1}^{D_z} \mathcal{N}(z_{ij}|0,1); \qquad p(\mathbf{x}_i|\mathbf{z}_i) = \prod_{j=1}^{D_x} p(x_{ij}|\mathbf{z}_i); \qquad p(t_i|\mathbf{z}_i) = \mathrm{Bern}(\sigma(f_1(\mathbf{z}_i))), \quad (2)$$

with $p(x_{ij}|\mathbf{z}_i)$ being an appropriate probability distribution for the covariate $j$ and $\sigma(\cdot)$ being the logistic function, $D_x$ the dimension of **x** and $D_z$ the dimension of **z**. For a continuous outcome we parametrize the probability distribution as a Gaussian with its mean given by a TARnet [48] architecture, i.e. a treatment specific function, and its variance fixed to $\hat{v}$, whereas for a discrete outcome we use a Bernoulli distribution similarly parametrized by a TARnet:

$$p(y_i|t_i, \mathbf{z}_i) = \mathcal{N}(\mu = \hat{\mu}_i, \sigma^2 = \hat{v}) \qquad \hat{\mu}_i = t_i f_2(\mathbf{z}_i) + (1 - t_i) f_3(\mathbf{z}_i) \qquad (3)$$

$$p(y_i|t_i, \mathbf{z}_i) = \mathrm{Bern}(\pi = \hat{\pi}_i) \qquad \hat{\pi}_i = \sigma(t_i f_2(\mathbf{z}_i) + (1 - t_i) f_3(\mathbf{z}_i)). \qquad (4)$$

Note that each of the $f_k(\cdot)$ is a neural network parametrized by its own parameters $\theta_k$ for $k = 1, 2, 3$. As we do not a-priori know the hidden confounder **z** we have to marginalize over it in order to learn the parameters of the model $\theta_k$. Since the non-linear neural network functions make inference intractable we will employ variational inference along with inference networks; these are neural networks that output the parameters of a fixed form posterior approximation over the latent variables

$\mathbf{z}$, e.g. a Gaussian, given the observed variables. By the definition of the model at Figure 1 we can see that the true posterior over $\mathbf{Z}$ depends on $\mathbf{X}, \mathbf{t}$ and $\mathbf{y}$. Therefore we employ the following posterior approximation:

$$q(\mathbf{z}_i|\mathbf{x}_i, t_i, y_i) = \prod_{j=1}^{D_z} \mathcal{N}(\mu_j = \bar{\mu}_{ij}, \sigma_j^2 = \bar{\sigma}_{ij}^2) \tag{5}$$

$$\bar{\boldsymbol{\mu}}_i = t_i \boldsymbol{\mu}_{t=0,i} + (1-t_i) \boldsymbol{\mu}_{t=1,i} \qquad \bar{\boldsymbol{\sigma}}_i^2 = t_i \boldsymbol{\sigma}_{t=0,i}^2 + (1-t_i) \boldsymbol{\sigma}_{t=1,i}^2$$

$$\boldsymbol{\mu}_{t=0,i}, \boldsymbol{\sigma}_{t=0,i}^2 = g_2 \circ g_1(\mathbf{x}_i, y_i) \qquad \boldsymbol{\mu}_{t=1,i}, \boldsymbol{\sigma}_{t=1,i}^2 = g_3 \circ g_1(\mathbf{x}_i, y_i),$$

where we similarly use a TARnet [48] architecture for the inference network, i.e. split them for each treatment group in $t$ after a shared representation $g_1(\mathbf{x}_i, y_i)$, and each $g_k(\cdot)$ is a neural network with variational parameters $\phi_k$. We can now form a single objective for the inference and model networks, the variational lower bound of this graphical model [27, 46]:

$$\mathcal{L} = \sum_{i=1}^{N} \mathbb{E}_{q(\mathbf{z}_i|\mathbf{x}_i, t_i, y_i)}[\log p(\mathbf{x}_i, t_i|\mathbf{z}_i) + \log p(y_i|t_i, \mathbf{z}_i) + \log p(\mathbf{z}_i) - \log q(\mathbf{z}_i|\mathbf{x}_i, t_i, y_i)]. \tag{6}$$

Notice that for out of sample predictions, i.e. new subjects, we require to know the treatment assignment $t$ along with its outcome $y$ before inferring the distribution over $\mathbf{z}$. For this reason we will introduce two auxiliary distributions that will help us predict $t_i, y_i$ for new samples. More specifically, we will employ the following distributions for the treatment assignment $t$ and outcomes $y$:

$$q(t_i|\mathbf{x}_i) = \text{Bern}(\pi = \sigma(g_4(\mathbf{x}_i))) \tag{7}$$

$$q(y_i|\mathbf{x}_i, t_i) = \mathcal{N}(\mu = \bar{\mu}_i, \sigma^2 = \bar{v}) \qquad \bar{\mu}_i = t_i(g_6 \circ g_5(\mathbf{x}_i)) + (1-t_i)(g_7 \circ g_5(\mathbf{x}_i)) \tag{8}$$

$$q(y_i|\mathbf{x}_i, t_i) = \text{Bern}(\pi = \bar{\pi}_i) \qquad \bar{\pi}_i = t_i(g_6 \circ g_5(\mathbf{x}_i)) + (1-t_i)(g_7 \circ g_5(\mathbf{x}_i)), \tag{9}$$

where we choose eq. 8 for continuous and eq. 9 for discrete outcomes. To estimate the parameters of these auxiliary distributions we will add two extra terms in the variational lower bound:

$$\mathcal{F}_{\text{CEVAE}} = \mathcal{L} + \sum_{i=1}^{N} \left( \log q(t_i = t_i^*|\mathbf{x}_i^*) + \log q(y_i = y_i^*|\mathbf{x}_i^*, t_i^*) \right), \tag{10}$$

with $\mathbf{x}_i, t_i^*, y_i^*$ being the observed values for the input, treatment and outcome random variables in the training set. We coin the name *Causal Effect Variational Autoencoder* (CEVAE) for our method.

## 4 Experiments

Evaluating causal inference methods is always challenging because we usually lack ground-truth for the causal effects. Common evaluation approaches include creating synthetic or semi-synthetic datasets, where real data is modified in a way that allows us to know the true causal effect or real-world data where a randomized experiment was conducted. Here we compare with two existing benchmark datasets where there is no need to model proxies, IHDP [21] and Jobs [33], often used for evaluating individual level causal inference. In order to specifically explore the role of proxy variables, we create a synthetic toy dataset, and introduce a new benchmark based on data of twin births and deaths in the USA.

For the implementation of our model we used Tensorflow [1] and Edward [52]. For the neural network architecture choices we closely followed [48]; unless otherwise specified we used 3 hidden layers with ELU [11] nonlinearities for the approximate posterior over the latent variables $q(\mathbf{Z}|\mathbf{X}, \mathbf{t}, \mathbf{y})$, the generative model $p(\mathbf{X}|\mathbf{Z})$ and the outcome models $p(\mathbf{y}|\mathbf{t}, \mathbf{Z}), q(\mathbf{y}|\mathbf{t}, \mathbf{X})$. For the treatment models $p(\mathbf{t}|\mathbf{Z}), q(\mathbf{t}|\mathbf{X})$ we used a single hidden layer neural network with ELU nonlinearities. Unless mentioned otherwise, we used a 20-dimensional latent variable $\mathbf{z}$ and used a small weight decay term for all of the parameters with $\lambda = .0001$. Optimization was done with Adamax [26] and a learning rate of $0.01$, which was annealed with an exponential decay schedule. We further performed early stopping according to the lower bound on a validation set. To compute the outcomes $p(\mathbf{y}|\mathbf{X}, do(\mathbf{t} = 1))$ and $p(\mathbf{y}|\mathbf{X}, do(\mathbf{t} = 0))$ we averaged over 100 samples from the approximate posterior $q(\mathbf{Z}|\mathbf{X}) = \sum_{\mathbf{t}} \int q(\mathbf{Z}|\mathbf{t}, \mathbf{y}, \mathbf{X})q(\mathbf{y}|\mathbf{t}, \mathbf{X})q(\mathbf{t}|\mathbf{X})d\mathbf{y}$.

Throughout this section we compare with several baseline methods. $LR1$ is logistic regression, $LR2$ is two separate logistic regressions fit to treated ($t = 1$) and control ($t = 0$). TARnet is a feed forward neural network architecture for causal inference [48].

## 4.1 Benchmark datasets

For the first benchmark task we consider estimating the individual and population causal effects on a benchmark dataset introduced by [21]; it is constructed from data obtained from the Infant Health and Development Program (IHDP). Briefly, the confounders $\mathbf{x}$ correspond to collected measurements of the children and their mothers used during a randomized experiment that studied the effect of home visits by specialists on future cognitive test scores. The treatment assignment is then "de-randomized" by removing from the treated set children with non-white mothers; for each unit a treated and a control outcome are then simulated, thus allowing us to know the "true" individual causal effects of the treatment. We follow [25, 48] and use 1000 replications of the simulated outcome, along with the same train/validation/testing splits. To measure the accuracy of the individual treatment effect estimation we use the Precision in Estimation of Heterogeneous Effect (PEHE) [21], PEHE $= \frac{1}{N}\sum_{i=1}^{N}((y_{i1} - y_{i0}) - (\hat{y}_{i1} - \hat{y}_{i0}))^2$, where $y_1, y_0$ correspond to the true outcomes under $t = 1$ and $t = 0$, respectively, and $\hat{y}_1, \hat{y}_0$ correspond to the outcomes estimated by our model. For the population causal effect we report the absolute error on the Average Treatment Effect (ATE). The results can be seen at Table 1. As we can see, CEVAE has decent performance, comparable to the Balancing Neural Network (BNN) of [25].

Table 1: Within-sample and out-of-sample mean and standard errors for the metrics for the various models at the IHDP dataset.

| Method | $\sqrt{\epsilon_{\text{PEHE}}^{\text{within-s.}}}$ | $\epsilon_{\text{ATE}}^{\text{within-s.}}$ | $\sqrt{\epsilon_{\text{PEHE}}^{\text{out-of-s.}}}$ | $\epsilon_{\text{ATE}}^{\text{out-of-s.}}$ |
|---|---|---|---|---|
| OLS-1 | 5.8±.3 | .73±.04 | 5.8±.3 | .94±.06 |
| OLS-2 | 2.4±.1 | .14±.01 | 2.5±.1 | .31±.02 |
| BLR | 5.8±.3 | .72±.04 | 5.8±.3 | .93±.05 |
| k-NN | 2.1±.1 | .14±.01 | 4.1±.2 | .79±.05 |
| TMLE | 5.0±.2 | .30±.01 | - | - |
| BART | 2.1±.1 | .23±.01 | 2.3±.1 | .34±.02 |
| RF | 4.2±.2 | .73±.05 | 6.6±.3 | .96±.06 |
| CF | 3.8±.2 | .18±.01 | 3.8±.2 | .40±.03 |
| BNN | 2.2±.1 | .37±.03 | 2.1±.1 | .42±.03 |
| CFRW | .71±.0 | .25±.01 | .76±.0 | .27±.01 |
| CEVAE | 2.7±.1 | .34±.01 | 2.6±.1 | .46±.02 |

Table 2: Within-sample and out-of-sample policy risk and error on the average treatment effect on the treated (ATT) for the various models on the Jobs dataset.

| Method | $R_{pol}^{\text{within-s.}}$ | $\epsilon_{\text{ATT}}^{\text{within-s.}}$ | $R_{pol}^{\text{out-of-s.}}$ | $\epsilon_{\text{ATT}}^{\text{out-of-s.}}$ |
|---|---|---|---|---|
| LR-1 | .22±.0 | .01±.00 | .23±.0 | .08±.04 |
| LR-2 | .21±.0 | .01±.01 | .24±.0 | .08±.03 |
| BLR | .22±.0 | .01±.01 | .25±.0 | .08±.03 |
| k-NN | .02±.0 | .21±.01 | .26±.0 | .13±.05 |
| TMLE | .22±.0 | .02±.01 | - | - |
| BART | .23±.0 | .02±.00 | .25±.0 | .08±.03 |
| RF | .23±.0 | .03±.01 | .28±.0 | .09±.04 |
| CF | .19±.0 | .03±.01 | .20±.0 | .07±.03 |
| BNN | .20±.0 | .04±.01 | .24±.0 | .09±.04 |
| CFRW | .17±.0 | .04±.01 | .21±.0 | .09±.03 |
| CEVAE | .15±.0 | .02±.01 | .26±.0 | .03±.01 |

For the second benchmark we consider the task described at [48] and follow closely their procedure. It uses a dataset obtained by the study of [33, 49], which concerns the effect of job training (treatment) on employment after training (outcome). Due to the fact that a part of the dataset comes from a randomized control trial we can estimate the "true" causal effect. Following [48] we report the absolute error on the Average Treatment effect on the Treated (ATT), which is the $\mathbb{E}\left[ITE(\mathbf{X})|\mathbf{t} = 1\right]$. For the individual causal effect we use the policy risk, that acts as a proxy to the individual treatment effect. The results after averaging over 10 train/validation/test splits can be seen at Table 2. As we can observe, CEVAE is competitive with the state-of-the art, while overall achieving the best estimate on the out-of-sample ATT.

## 4.2 Synthetic experiment on toy data

To illustrate that our model better handles hidden confounders we experiment on a toy simulated dataset where the marginal distribution of $\mathbf{X}$ is a mixture of Gaussians, with the hidden variable $\mathbf{Z}$ determining the mixture component. We generate synthetic data by the following process:

$$\mathbf{z}_i \sim \text{Bern}(0.5); \qquad \mathbf{x}_i|\mathbf{z}_i \sim \mathcal{N}\left(\mathbf{z}_i, \sigma_{z_1}^2 \mathbf{z}_i + \sigma_{z_0}^2(1 - \mathbf{z}_i)\right)$$
$$\mathbf{t}_i|\mathbf{z}_i \sim \text{Bern}(0.75\mathbf{z}_i + 0.25(1 - \mathbf{z}_i)); \qquad \mathbf{y}_i|\mathbf{t}_i, \mathbf{z}_i \sim \text{Bern}\left(\text{Sigmoid}\left(3(\mathbf{z}_i + 2(2\mathbf{t}_i - 1)))\right)\right), \tag{11}$$

where $\sigma_{z_0} = 3$, $\sigma_{z_1} = 5$ and Sigmoid is the logistic sigmoid function. This generation process introduces hidden confounding between $\mathbf{t}$ and $\mathbf{y}$ as $\mathbf{t}$ and $\mathbf{y}$ depend on the mixture assignment $\mathbf{z}$ for $\mathbf{x}$. Since there is significant overlap between the two Gaussian mixture components we expect that methods which do not model the hidden confounder $z$ will not produce accurate estimates for the treatment effects. We experiment with both a binary $z$ for CEVAE, which is close to the true

model, as well as a 5-dimensional continuous $z$ in order to investigate the robustness of CEVAE w.r.t. model misspecification. We evaluate across samples size $N \in \{1000, 3000, 5000, 10000, 30000\}$ and provide the results in Figure 3. We see that no matter how many samples are given, $LR1$, $LR2$ and TARnet are not able to improve their error in estimating ATE directly from the proxies. On the other hand, CEVAE achieves significantly less error. When the latent model is correctly specified (CEVAE bin) we do better even with a small sample size; when it is not (CEVAE cont) we require more samples for the latent space to imitate more closely the true binary latent variable.

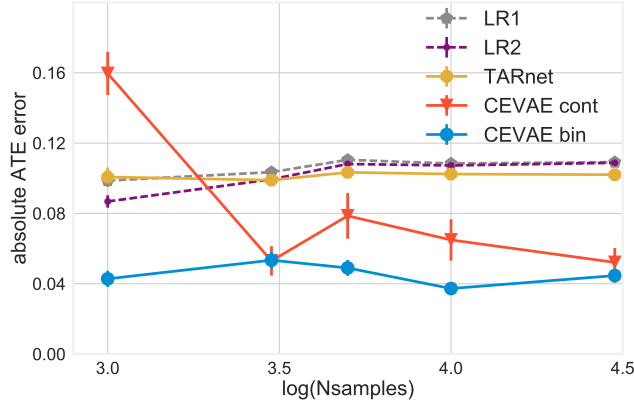

Figure 3: Absolute error of estimating ATE on samples from the generative process (11). CEVAE bin and CEVAE cont are CEVAE with respectively binary or continuous 5-dim latent $z$. See text above for description of the other methods.

### 4.3  Binary treatment outcome on Twins

We introduce a new benchmark task that utilizes data from twin births in the USA between 1989-1991 [3] [3]. The treatment $\mathbf{t} = 1$ is being born the heavier twin whereas, the outcome corresponds to the mortality of each of the twins in their first year of life. Since we have records for both twins, their outcomes could be considered as the two potential outcomes with respect to the treatment of being born heavier. We only chose twins which are the same sex. Since the outcome is thankfully quite rare (3.5% first-year mortality), we further focused on twins such that both were born weighing less than $2kg$. We thus have a dataset of 11984 pairs of twins. The mortality rate for the lighter twin is 18.9%, and for the heavier 16.4%, for an average treatment effect of $-2.5\%$. For each twin-pair we obtained 46 covariates relating to the parents, the pregnancy and birth: mother and father education, marital status, race and residence; number of previous births; pregnancy risk factors such as diabetes, renal disease, smoking and alcohol use; quality of care during pregnancy; whether the birth was at a hospital, clinic or home; and number of gestation weeks prior to birth.

In this setting, for each twin pair we observed both the case $\mathbf{t} = 0$ (lighter twin) and $\mathbf{t} = 1$ (heavier twin). In order to simulate an observational study, we selectively hide one of the two twins; if we were to choose at random this would be akin to a randomized trial. In order to simulate the case of hidden confounding with proxies, we based the treatment assignment on a single variable which is highly correlated with the outcome: GESTAT10, the number of gestation weeks prior to birth. It is ordinal with values from 0 to 9 indicating birth before 20 weeks gestation, birth after 20-27 weeks of gestation and so on [4]. We then set $\mathbf{t}_i | \mathbf{x}_i, \mathbf{z}_i \sim \text{Bern}\left(\sigma(w_o^\top \mathbf{x} + w_h(\mathbf{z}/10 - 0.1))\right)$, $w_o \sim \mathcal{N}(0, 0.1 \cdot I)$, $w_h \sim \mathcal{N}(5, 0.1)$, where $\mathbf{z}$ is GESTAT10 and $\mathbf{x}$ are the 45 other features.

We created proxies for the hidden confounder as follows: We coded the 10 GESTAT10 categories with one-hot encoding, replicated 3 times. We then randomly and independently flipped each of these 30 bits. We varied the probabilities of flipping from 0.05 to 0.5, the latter indicating there is no direct information about the confounder. We chose three replications following the well-known result that three independent views of a latent feature are what is needed to guarantee that it can be recovered

[30, 2, 5]. We note that there might still be proxies for the confounder in the other variables, such as the incompetent cervix covariate which is a known risk factor for early birth. Having created the dataset, we focus our attention on two tasks: Inferring the mortality of the unobserved twin (counterfactual), and inferring the average treatment effect. We compare with TARnet, LR1 and LR2. We vary the number of hidden layers for TARnet and CEVAE (*nh* in the figures). We note that while TARnet with 0 hidden layers is equivalent to $LR2$, CEVAE with 0 hidden layers still infers a latent space and is thus different. The results are given respectively in Figures 4(a) (higher is better) and 4(b) (lower is better).

For the counterfactual task, we see that for small proxy noise all methods perform similarly. This is probably due to the gestation length feature being very informative; for $LR1$, the noisy codings of this feature form 6 of the top 10 most predictive features for mortality, the others being sex (males are more at risk), and 3 risk factors: incompetent cervix, mother lung disease, and abnormal amniotic fluid. For higher noise, TARnet, $LR1$ and $LR2$ see roughly similar degradation in performance; CEVAE, on the other hand, is much more robust to increasing proxy noise because of its ability to infer a cleaner latent state from the noisy proxies. Of particular interest is CEVAE $nh = 0$, which does much better for counterfactual inference than the equivalent $LR2$, probably because $LR2$ is forced to rely directly on the noisy proxies instead of the inferred latent state. For inference of average treatment effect, we see that at the low noise levels CEVAE does slightly worse than the other methods, with CEVAE $nh = 0$ doing noticeably worse. However, similar to the counterfactual case, CEVAE is significantly more robust to proxy noise, achieving quite a low error even when the direct proxies are completely useless at noise level $0.5$.

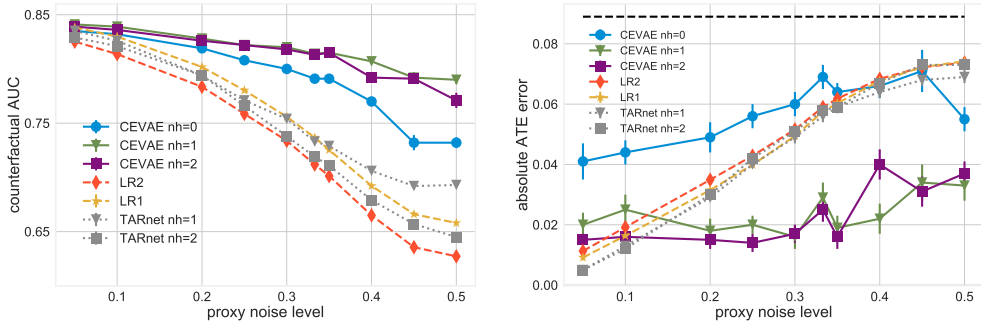

(a) Area under the curve (AUC) for predicting the mortality of the unobserved twin in a hidden confounding experiment; higher is better.

(b) Absolute error ATE estimate; lower is better. Dashed black line indicates the error of using the naive ATE estimator: the difference between the average treated and average control outcomes.

Figure 4: Results on the Twins dataset. $LR1$ is logistic regression, $LR2$ is two separate logistic regressions fit on the treated and control. "nh" is number of hidden layers used. TARnet with $nh = 0$ is identical to $LR2$ and not shown, whereas CEVAE with $nh = 0$ has a latent space component.

## 5  Conclusion

In this paper we draw a connection between causal inference with proxy variables and the groundbreaking work in the machine learning community on latent variable models. Since almost all observational studies rely on proxy variables, this connection is highly relevant.

We introduce a model which is the first attempt at tying these two ideas together: The Causal Effect Variational Autoencoder (CEVAE), a neural network latent variable model used for estimating individual and population causal effects. In extensive experiments we showed that it is competitive with the state-of-the art on benchmark datasets, and more robust to hidden confounding both at a toy artificial dataset as well as modifications of real datasets, such as the newly introduced Twins dataset. For future work, we plan to employ the expanding set of tools available for latent variables models (e.g. Kingma et al. [28], Tran et al. [51], Maaløe et al. [35], Ranganath et al. [44]), as well as to further explore connections between method of moments approaches such as Anandkumar et al. [5] with the methods for effect restoration given by Kuroki and Pearl [32], Miao et al. [37].

## Acknowledgements

We would like to thank Fredrik D. Johansson for valuable discussions, feedback and for providing the data for IHDP and Jobs. We would also like to thank Maggie Makar for helping with the Twins dataset. Christos Louizos and Max Welling were supported by TNO, NWO and Google. Joris Mooij was supported by the European Research Council (ERC) under the European Union's Horizon 2020 research and innovation programme (grant agreement 639466).

## Footnotes

*Canadian Institute For Advanced Research

[2]Including observed covariates which do not affect the intervention or outcome, and therefore are not truly confounders.

[3]Data taken from the denominator file at http://www.nber.org/data/linked-birth-infant-death-data-vital-statistics-data.html

[4]The partition is given in the original dataset from NBER.

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
