[Supplementary Material]

# Causal Effect Inference with Deep Latent-Variable Models

**Christos Louizos**
University of Amsterdam
TNO Intelligent Imaging
c.louizos@uva.nl

**Uri Shalit**
New York University
CIMS
uas1@nyu.edu

**Joris Mooij**
University of Amsterdam
j.m.mooij@uva.nl

**David Sontag**
Massachusetts Institute of Technology
CSAIL & IMES
dsontag@csail.mit.edu

**Richard Zemel**
University of Toronto
CIFAR[*]
zemel@cs.toronto.edu

**Max Welling**
University of Amsterdam
CIFAR[*]
m.welling@uva.nl

## Appendix

### A. Simple example where one should not adjust for proxy variables

Let $\mathbf{Z}, \mathbf{X}, \mathbf{t}, \mathbf{y}$ all be binary variables following the graphical model in Figure 1(a). Assume the following model:

1. $p(\mathbf{Z} = 1) = p(\mathbf{Z} = 0) = 0.5$.

2. $p(\mathbf{X} = 1|\mathbf{Z} = 1) = p(\mathbf{X} = 0|\mathbf{Z} = 0) = \rho_x$.

3. $p(\mathbf{t} = 1|\mathbf{Z} = 1) = p(\mathbf{t} = 0|\mathbf{Z} = 0) = \rho_t$.

4. $\mathbf{y} = \mathbf{t} \oplus \mathbf{Z}$ ($\mathbf{y}$ is deterministic)

We will look at the quantity $p(\mathbf{y}|do(\mathbf{t} = 1)$ and show that one should not simply adjust for $\mathbf{X}$ when computing it. We have the following identities:

$$
\begin{aligned}
&p(\mathbf{y} = 1|do(\mathbf{t} = 1)) = \\
&\sum_{\mathbf{z}} p(\mathbf{y} = 1|do(\mathbf{t} = 1), \mathbf{Z} = \mathbf{z})p(\mathbf{Z} = \mathbf{z}|do(\mathbf{t} = 1)) = \\
&\sum_{\mathbf{z}} p(\mathbf{y} = 1|\mathbf{t} = 1, \mathbf{Z} = \mathbf{z})p(\mathbf{Z} = \mathbf{z}|\mathbf{t} = 1) = \\
&\sum_{\mathbf{z}} p(\mathbf{y} = 1|\mathbf{t} = 1, \mathbf{Z} = \mathbf{z})p(\mathbf{Z} = \mathbf{z}) = \\
&p(\mathbf{z} = 0) = 0.5.
\end{aligned}
$$

---

[*]Canadian Institute For Advanced Research

The covariate adjustment formula if we treated $\mathbf{X}$ as if it's the only confounder:

$$p^{\text{wrong}}(\mathbf{y} = 1 | do(\mathbf{t} = 1)) =$$

$$\sum_{\mathbf{x}} p(\mathbf{y} = 1 | \mathbf{t} = 1, \mathbf{X} = \mathbf{x}) p(\mathbf{X} = \mathbf{x}) =$$

$$0.5 \cdot (p(\mathbf{y} = 1 | \mathbf{t} = 1, \mathbf{x} = 0) + p(\mathbf{y} = 1 | \mathbf{t} = 1, \mathbf{x} = 1)) =$$

$$0.5 \cdot \frac{p(\mathbf{t} = 1 | \mathbf{z} = 0) p(\mathbf{x} = 0 | \mathbf{z} = 0) p(\mathbf{z} = 0)}{p(\mathbf{t} = 1, \mathbf{x} = 0 | \mathbf{z} = 0) p(\mathbf{z} = 0) + p(\mathbf{t} = 1, \mathbf{x} = 0 | \mathbf{z} = 1) p(\mathbf{z} = 1)} +$$

$$0.5 \cdot \frac{p(\mathbf{t} = 1 | \mathbf{z} = 0) p(\mathbf{x} = 1 | \mathbf{z} = 0) p(\mathbf{z} = 0)}{p(\mathbf{t} = 1, \mathbf{x} = 1 | \mathbf{z} = 0) p(\mathbf{z} = 0) + p(\mathbf{t} = 1, \mathbf{x} = 1 | \mathbf{z} = 1) p(\mathbf{z} = 1)} =$$

$$0.5 \cdot \left( \frac{(1 - \rho_t) \rho_x}{(1 - \rho_t) \rho_x + \rho_t (1 - \rho_x)} + \frac{(1 - \rho_t)(1 - \rho_x)}{(1 - \rho_t)(1 - \rho_x) + \rho_t \rho_x} \right).$$

A short inspection shows that we obtain the correct answer, $p^{\text{wrong}}(\mathbf{y} = 1 | do(\mathbf{t} = 1)) = 0.5$, exactly under one of the following two conditions:

1. $\rho_t = 0.5$, i.e. treatment is assigned randomly.
2. $\rho_x = 0$ or $\rho_x = 1$, i.e. $\mathbf{X}$ is exactly equal to $\mathbf{Z}$ or $1 - \mathbf{Z}$, and thus is a perfect proxy for $\mathbf{Z}$.

The crucial misstep in $p^{\text{wrong}}$ above is the fact that $p(\mathbf{y} = 1 | do(\mathbf{t} = 1), \mathbf{x}) \neq p(\mathbf{y} = 1 | \mathbf{t} = 1, \mathbf{x})$, while on the other hand $p(\mathbf{y} = 1 | do(\mathbf{t} = 1), \mathbf{z}) = p(\mathbf{y} = 1 | \mathbf{t} = 1, \mathbf{z})$.

We note that because of the symmetry in the conditional distributions $p(\mathbf{X} | \mathbf{Z} = 1)$ and $p(\mathbf{X} | \mathbf{Z} = 0)$, we will actually have that:

$$p^{\text{wrong}}(\mathbf{y} = 1 | do(\mathbf{t} = 1)) - p^{\text{wrong}}(\mathbf{y} = 1 | do(\mathbf{t} = 0)) = p(\mathbf{y} = 1 | do(\mathbf{t} = 1)) - p(\mathbf{y} = 1 | do(\mathbf{t} = 0)).$$

It is straightforward to show that this situation does not happen once we set different values for the two conditional distributions of $p(\mathbf{X} | \mathbf{Z} = 1)$ and $p(\mathbf{X} | \mathbf{Z} = 0)$, in which case both $p^{\text{wrong}}(\mathbf{y} = 1 | do(\mathbf{t})) \neq p(\mathbf{y} = 1 | do(\mathbf{t}))$ for $\mathbf{t} = 0, 1$, and:

$$p^{\text{wrong}}(\mathbf{y} = 1 | do(\mathbf{t} = 1)) - p^{\text{wrong}}(\mathbf{y} = 1 | do(\mathbf{t} = 0)) \neq p(\mathbf{y} = 1 | do(\mathbf{t} = 1)) - p(\mathbf{y} = 1 | do(\mathbf{t} = 0)).$$