[Reviews · NeurIPS 2017]

Reviewer 1



This paper tackles the problem of estimating individual treatment effects using proxy variables. Proxy variables partially capture the effects of the unobserved confounders, but not all effects. Therefore, simply conditioning on proxy variables does not work; one needs a better way of controlling for unobserved confounding. The authors present a method based on variational auto-encoders that models the joint probability of observed and unobserved variables. They are able to model a multi-dimensional representation for the confounders (z). Their identification still depends on some parametric assumptions about z (e.g., normally distributed), but it is still an improvement over past work because they are able to model z as high-dimensional, while also allowing high dimensional proxy variables. I like how the authors tested their method on multiple real and simulated datasets. While any causal inference method is unlikely to be optimal for all contexts, the results do show encouraging results. From simulations, it seems that this result is able to perform well even as the noise in proxy variables is increased, although more testing is needed to understand the properties of the method. This is one of the weaknesses of the proposed approach; hard to characterize any properties of the identification procedure because we do not have a good theory for variational autoencoders. Still, the proposed method is flexible enough that it can be practically useful to control for non-trivial forms of confounding in individual causal effects.

Reviewer 2



- The authors utilized Variational Autoencoders (VAE) in order to estimate the individual treatment effect (ITE) and the average treatment effect (ATE) for the causal model in Figure 1. However, five other causal diagrams can also be considered with confounder proxies (see Figure 1 in [Miao et al., 2016]). It would be interesting to check whether the proposed method can be applied to all these causal diagrams or there are some limitations. - Please give some justifications why parametrizing the probability distributions in the form of Equations (2-4) would be a good choice for estimating ITE and ATE. For instance, why covariates of x_i are independent given z_i? Is it restricting the expressive power of this method? - In your method, do we need to know the dimension of latent variable z? For instance, in Experimental Results section, it is assumed that z is a 20-dimensional variable. Does the performance degrade if D_z is not adjusted correctly? - In Table 1, is there any good reason that BNN has a better performance than CEVAE? - When z is categorical, it seems that we can just estimate p(x_i|z_i) by your method and then estimate ITE and ATE from the results in [Kuroki and Pearl, 2011]. It would be interesting to compare the performance of this method with CAVAE? - Figure 3 shows that model misspecification may increase the sample complexity. It would be interesting to characterize the sample complexity overhead from model misspecification. - Under some conditions, can we have a theoretical guarantee that CEVAE gives a good approximation of ITE or ATE? - Some typos: o Line 126: p(Z,Xt,y)->p(Z,X,t,y) o In equation (2), please define D_x and D_z.

Reviewer 3



This paper addresses the problem of predicting the individual treatment effects of a policy intervention on some variable, given a set of proxy variables used to attempt to capture the unknown latent variable space. The approach is based on variational autoencoders (VAE) to leverage big data for causal inference. VAEs have the advantage of making few assumptions about the latent structure and strong empirical performance, but the disadvantage of lacking a guarantee to converge to the true minimum. Empirical results indicate the proposed method compares favorably to some benchmarks. In general the paper is well written and the results appear sound and consistent with the existing literature. However, there doesn't seem to be a lot of novel content in the paper: 1) On the causality side, the contribution seems to be theorem 1, but this is basically a restatement of a do-calculus result from (Pearl, 2009). 2) On the deep learning side, the VAE approach seems to be mostly based on TARnet from (Shalit et al., 2016) [more below]. The paper seems to basically be a combination of these two methods. In section 3, it's not clear what the main differences are between the proposed CEVAE method and the TARnet method from (Shalit et al., 2016) - because of this, it's difficult to determine whether there is any significantly novelty here and whether this might be relevant to the potential impact of the paper. Unless I have misunderstood section 3, there does not seem to be enough novelty to warrant acceptance for this paper.